# Diagonal Denoising for Spatially Correlated Noise Based on Diagonalization Decorrelation in Underwater Radiated Noise Measurement

**Guoqing Jiang** [1,2], **Chao Sun** [1,2,*] **and Lei Xie** [1,2]

1   School of Marine Science and Technology, Northwestern Polytechnical University, Xi'an 710072, China; jiangguoqing@mail.nwpu.edu.cn (G.J.); xielei2014@mail.nwpu.edu.cn (L.X.)
2   Shaanxi Key Laboratory of Underwater Information Technology, Northwestern Polytechnical University, Xi'an 710072, China
*   Correspondence: csun@nwpu.edu.cn

**Abstract:** In underwater radiated noise measurement using a vertical linear array, a diagonalization-decorrelation-based diagonal denoising method is proposed to improve the denoising effect for spatially correlated noise. Firstly, the ambient noise cross-spectral matrix is measured without the radiated noise source. Subsequently, the eigenvector matrix of the ambient noise cross-spectral matrix is utilized to implement a unitary transformation for the received data, which eliminates the correlation of the received noise and transforms the received noise cross-spectral matrix into a diagonal matrix, then the noise components are removed by diagonal denoising. Finally, the denoised cross-spectral matrix is used to estimate the power of the radiated noise by beamforming. Consequently, the influence of spatially correlated noise on radiated noise measurement is reduced. The effectiveness of the proposed method is validated and compared with the diagonal denoising method and the whitening-decorrelation-based diagonal denoising method via numerical simulations and experimental data. Under the ideal condition, the noise reduction performances of the proposed method and the whitening-decorrelation-based diagonal denoising method are equal and better than that of the diagonal denoising method. In practice, the number of snapshots is limited, so there is an inevitable mismatch between the ambient noise cross-spectral matrix and the received noise cross-spectral matrix due to the randomness of noise. The mismatch results in imperfect whitening and diagonalization, which reduces the denoising effect. However, the simulation results indicate that the proposed method still reduces more correlated noise and has a better performance on underwater radiated noise measurement compared with the diagonal denoising method and the whitening-decorrelation-based diagonal denoising method even if the number of snapshots is finite.

**Keywords:** underwater radiated noise measurement; diagonal denoising; diagonalization; decorrelation; spatially correlated noise

## 1. Introduction

Underwater radiated noise measurement is used for quantification and qualification of an underwater radiated noise source, and the resulting quantities are based on the root-mean-square Sound Pressure Level (SPL) [1]. The underwater SPL measurements are performed at a distance and then adjusted to the 1 m normalized distance. Underwater radiated noise measurement is essential to monitor the anthropogenic noise in the ocean [2–5], such as offshore wind farms and all kinds of vessels. The common radiated noise measurement approach is to use a single hydrophone [6–8]. The influence of ambient noise is negligible in the high Signal-to-Noise Ratio (SNR) case and the measured SPL is closer to the true value. However, the growing noise level leads to the declining SNR of the received data and eventually decreases the accuracy of radiated noise measurement [9,10]. To solve this problem, several studies have proposed using a Vertical Linear Array (VLA) to receive the signal and suppress the ambient noise by beamforming [11–13]. However, when

the input SNR of the beamformer is too low, the output SNR of it with a small aperture array will still be low, which may not ensure a good measurement result. In this case, some denoising processing is necessary to reduce the ambient noise before beamforming.

Many denoising algorithms have been developed in different applications [14–18]. For radiated noise measurement, the requirement that the signal power should not be removed during the denoising process is essential to ensure that the measured SPL is not underestimated. The array received spatially uncorrelated noise mainly contributes to the diagonal elements of the Cross-Spectral Matrix (CSM) when the number of snapshots is sufficient. According to the structure of noise CSM that the noise component is concentrated on its diagonal, a simple denoising algorithm is to subtract a constant from all diagonal elements of the received data CSM to reduce the noise component. When the noise powers received by different hydrophones are not equal, the diagonal elements are different and thus this denoising algorithm only reduces parts of the noise. The robust principal component analysis method can decompose the received data CSM into the signal CSM and the noise CSM by taking advantage of both the low-rank property of the signal CSM and the sparse property of the noise CSM, thereby removing the influence of noise [15,16]. But the robust principal component analysis is computationally intensive, and it is difficult to choose the normalization parameter for optimal denoising performance. Jiang et al. proposed a Diagonal Denoising (DD) algorithm that reduces the diagonal noise elements as much as possible until the conventional beamforming output spatial spectrum becomes sparsest [17]. However, the output spatial spectrum level is lower than the real spectrum of the signal because extra power is removed in achieving the sparsest beamforming output spatial spectrum. Therefore, the denoising algorithm is not suitable for radiated noise measurement. Hald et al. proposed a novel DD algorithm whose denoising factors are different from each other, which provides a better denoising effect because the different denoising factors can reduce the diagonal elements as much as possible until the denoised matrix to be positive semidefinite [18]. In this method, the denoising factors are obtained by maximizing the sum of all denoising factors and maintaining the denoised matrix to be positive semidefinite. The solving process is stable, convergent, and computationally efficient. However, the received noise CSM is no longer a diagonal matrix when spatially correlated noise is included in the received noise, leading to the denoising effects of all above-mentioned denoising algorithms degrading. For this reason, the received data needs some decorrelation processing by taking advantage of the statistical property of the stationary noise, such as CSM, to transform the received noise CSM into a diagonal matrix.

Whitening is a common tool that transforms the received noise CSM into an identity matrix [19], then the noise can be removed by subtracting an identity matrix from the whitened matrix, named the Whitening-Decorrelation-based DD (WD-DD) method. In practice, there is a mismatch between the ambient noise CSM and the received noise CSM when the number of snapshots is limited. Unfortunately, whitening is sensitive because the inversion operation in whitening will magnify the mismatch error, which results in imperfect whitening, thereby reducing the denoising effect. To improve the denoising effect for spatially correlated noise, we propose a Diagonalization-Decorrelation-based DD (DD-DD) method. The accuracy of underwater radiated noise measurement, of course, is improved because of the increasing denoising effect. The received noise CSM is transformed into a diagonal matrix by a unitary transformation, and then the noise components in the diagonalized matrix are removed by DD. The denoising effects of DD-DD, WD-DD, and DD are simulated under the ideal condition and the realistic condition, respectively. Then, the influences of denoising on radiated noise measurement are simulated with the SNR, the number of snapshots, and the frequency of the received data, respectively. In addition, the effectiveness of DD-DD on underwater radiated noise measurement is also validated via experimental data.

The rest of the paper is organized as follows. Section 2 introduces the receiving data model with a VLA and the underwater radiated noise measurement approach based on beamforming. Section 3 describes the DD algorithm for spatially uncorrelated noise and the proposed DD-DD method for spatially correlated noise. In Section 4, the denoising

effect and the influence on radiated noise measurement of DD-DD are demonstrated and compared with DD and WD-DD via numerical simulations. Section 5 further verifies the validity of DD-DD in practice by experimental data. Finally, Section 6 provides a summary.

## 2. Radiated Noise Measurement Using a Vertical Linear Array

### 2.1. Data Model

A VLA consisting of $M$ equally spaced hydrophones is used to receive the noise radiated from a point source. The received data at the $m$th hydrophone at time instant $t$, $x_m(t)$, can be formulated as:

$$x_m(t) = g_m(t) * s(t) + n_m(t), \quad m = 1, 2, \ldots, M, \tag{1}$$

where $*$ denotes the convolution operator, $s(t)$ is the radiated noise, $n_m(t)$ is the additive noise received by the $m$th hydrophone, and $g_m(t)$ is the impulse response of the channel from the source to the $m$th hydrophone. Because the broadband signal is usually processed with beamforming in the frequency-domain, the received data of the array are converted to the frequency-domain by Short-Time Fourier Transform (STFT) and expressed as:

$$X(n, f) = G(n, f)S(n, f) + N(n, f), \tag{2}$$

where $n$ is the block index in time and $f$ is the frequency. The vectors are defined as:

$$X(n, f) = [X_1(n, f), X_2(n, f), \ldots, X_M(n, f)]^T \in \mathbb{C}^{M \times 1},$$
$$G(n, f) = [G_1(n, f), G_2(n, f), \ldots, G_M(n, f)]^T \in \mathbb{C}^{M \times 1},$$
$$N(n, f) = [N_1(n, f), N_2(n, f), \ldots, N_M(n, f)]^T \in \mathbb{C}^{M \times 1}$$

with the superscript T denoting the transpose, and $X_m(n, f)$, $G_m(n, f)$, $S(n, f)$, and $N_m(n, f)$ are the STFTs of $x_m(t)$, $g_m(t)$, $s(t)$, and $n_m(t)$, respectively.

### 2.2. Radiated Noise Measurement Based on Beamforming

For underwater radiated noise measurement, the relative position of the source and VLA is known. Hence, the near-field conventional beamforming focused on the source in the direction of the direct path is applied to the array of received data to suppress the ambient noise, and the beamformer output is expressed as:

$$Y(n, f) = w(f)^H X(n, f) = w(f)^H G(n, f)S(n, f) + w(f)^H N(n, f), \tag{3}$$

where the superscript H denotes the Hermitian transpose, $w(f)$ is the weighting vector and is expressed as:

$$w(f) = \frac{1}{M} \left( \frac{r_1}{r_0} e^{-jk(r_1 - r_0)}, \frac{r_2}{r_0} e^{-jk(r_2 - r_0)}, \ldots, \frac{r_M}{r_0} e^{-jk(r_M - r_0)} \right)^T, \tag{4}$$

where j denotes imaginary unit, $k = 2\pi f / c$ represents the wavenumber with $c$ denoting sound speed, $r_m$ is the distance between the $m$th hydrophone and the source, and $r_0$ denotes the distance between the reference hydrophone and the source. For simplicity, $n$ and $f$ will be omitted in the following expressions.

The Power Spectral Density (PSD) of the beamformer output $Y$ at the frequency $f$ can be formulated in decibels [20]

$$\begin{aligned} P_Y &= 10 \lg \left( \frac{|Y|^2}{L f_s} \right) = 10 \lg \left( \frac{w^H R w}{L f_s} \right) \\ &\approx 10 \lg \left( \frac{|w^H GS|^2}{L f_s} \right) \\ &= P_S + 10 \lg |w^H G|^2, \end{aligned} \tag{5}$$

where $L$ is the length of the window function used in STFT, $f_s$ is the sampling frequency, $\boldsymbol{R} = \mathbb{E}(\boldsymbol{X}\boldsymbol{X}^{\mathrm{H}})$ denotes the CSM of the array received data with $\mathbb{E}(\cdot)$ representing the expectation, and $P_S = 10\lg(|S|^2/(Lf_s))$ denoting the PSD of the radiated noise. The power of ambient noise is negligible compared with that of the radiated noise when the SNR at the beamformer output is high. So the approximation in Equation (5) is reasonable with high SNR of beamformer output.

From Equation (5), the PSD of the radiated noise can be estimated

$$\hat{P}_S \triangleq 10\lg\left(\frac{\boldsymbol{w}^{\mathrm{H}}\boldsymbol{R}\boldsymbol{w}}{Lf_s}\right) - 10\lg|\boldsymbol{w}^{\mathrm{H}}\boldsymbol{G}|^2, \tag{6}$$

where $CF = -10\lg|\boldsymbol{w}^{\mathrm{H}}\boldsymbol{G}|^2$ is a correction factor. The correction factor can be decomposed into

$$CF = -10\lg\left(\frac{\boldsymbol{G}^{\mathrm{H}}\boldsymbol{G}}{M}\right) - 10\lg\left(\frac{|\boldsymbol{w}^{\mathrm{H}}\boldsymbol{G}|^2}{\frac{\boldsymbol{G}^{\mathrm{H}}\boldsymbol{G}}{M}}\right), \tag{7}$$

where the first item and the second item denote the mean transmission loss of $M$ hydrophones and the signal gain of the beamformer, respectively. Therefore, the correction factor has two purposes, one is compensating the transmission loss for normalizing the PSD to the distance of 1 m, the other is eliminating the beamforming effect on the PSD of the signal. For underwater radiated noise measurement, the experiment can be performed in a specific area where environment parameters can be obtained precisely. Then the transfer functions $\boldsymbol{G}$ can be estimated by the KrakenC model of the Acoustics Toolbox and the correction factor can be calculated [21].

In Equation (6), the received data CSM $\boldsymbol{R}$ contains the power of both the radiated noise and the ambient noise. The influence of ambient noise can be ignored only if the SNR at the beamformer output is high (generally greater than 10 dB). But in the low SNR case, the ambient noise power is comparable to that of the radiated noise, omitting the ambient noise power leads to a great error in the underwater radiated noise measurement. In this case, some denoising processing is necessary to reduce the ambient noise components in $\boldsymbol{R}$, and consequently improve the accuracy of the measurement. Note that the denoising processing is applied to the received data CSM before beamforming, so it only improves the input SNR of the beamformer and does not affect beamforming.

## 3. Denoising Methods

Assuming the radiated noise and the ambient noise are uncorrelated, then the ideal CSM of the received data can be formulated as

$$\boldsymbol{R} = \mathbb{E}(\boldsymbol{X}\boldsymbol{X}^{\mathrm{H}}) = \boldsymbol{R}_s + \boldsymbol{R}_n, \tag{8}$$

where $\boldsymbol{R}_s = \sigma_s^2 \boldsymbol{G}\boldsymbol{G}^{\mathrm{H}}$ is the received signal CSM with $\sigma_s^2 = \mathbb{E}(SS^{\mathrm{H}})$ denoting the power of the radiated noise and $\boldsymbol{R}_s$ is positive semidefinite, $\boldsymbol{R}_n = \mathbb{E}(\boldsymbol{N}\boldsymbol{N}^{\mathrm{H}})$ is the received noise CSM.

### 3.1. Diagonal Denoising for Spatially Uncorrelated Noise

In the spatially uncorrelated noise field, the received noise CSM $\boldsymbol{R}_n$ is a diagonal matrix. The powers of the noise received by different hydrophones are not equal when the noise field is inhomogeneous, yielding different diagonal elements in $\boldsymbol{R}_n$. To suppress noise as much as possible in this situation, the DD algorithm with different denoising factors is adopted [18] and formulated as

$$\begin{aligned} \max_{\boldsymbol{d}} \quad & \sum_{m=1}^{M} d_m, \\ \text{subject to} \quad & \boldsymbol{R}_{DD} = \boldsymbol{R} - \boldsymbol{D} \geq 0, \end{aligned} \tag{9}$$

where $d_m, m = 1, 2, \ldots, M$ is the denoising factor for the $m$th diagonal element of $\boldsymbol{R}$, $\boldsymbol{D} = \mathrm{diag}(\boldsymbol{d})$ is the denoising matrix with $\boldsymbol{d} = (d_1, d_2, \ldots, d_M)^{\mathrm{T}}$ and $\mathrm{diag}(\cdot)$ denoting a diagonal matrix formed by a vector, $\boldsymbol{R}_{DD}$ is the denoised CSM and $\boldsymbol{R}_{DD} \geq 0$ implies that $\boldsymbol{R}_{DD}$ is positive semidefinite.

The convex optimal problem expressed by Equation (9) can be solved easily by the CVX toolbox with Matlab [22,23]. The constraint that the denoised CSM is positive semidefinite is used to determine the upper limits of the denoising factors because the signal CSM $\boldsymbol{R}_s$ is positive semidefinite. As mentioned in Section 1, it is important for radiated noise measurement to ensure that the signal power is not removed during the denoising process. Hence, in Appendix A, we have proven that the maximum of $d_m$ does not exceed the diagonal elements of $\boldsymbol{R}_n$, which guarantees that DD would not reduce the signal power.

The estimated PSD of the radiated noise is calculated by replacing $\boldsymbol{R}$ in Equation (6) with the denoised CSM $\boldsymbol{R}_{DD}$

$$\hat{P}_S = 10 \log \left( \frac{\boldsymbol{w}^{\mathrm{H}} \boldsymbol{R}_{DD} \boldsymbol{w}}{L f_s} \right) - 10 \lg |\boldsymbol{w}^{\mathrm{H}} \boldsymbol{G}|^2. \tag{10}$$

The ambient noise in $\boldsymbol{R}_{DD}$ has been reduced, thus the influence of noise on the radiated noise measurement is decreased and the estimated PSD is more accurate.

### 3.2. Diagonalization-Decorrelation-Based Diagonal Denoising for Spatially Correlated Noise

In some cases, the spatially correlated noise is dominant in the ambient noise, resulting in a non-diagonal noise CSM $\boldsymbol{R}_n$. Hence, the performance of the DD algorithm degrades. To solve this problem, we propose a diagonalization-decorrelation-based diagonal denoising method. In this method, the spatial decorrelation processing is applied for the array received noise, so the noise CSM is diagonalized. Consequently, the noise components can be removed by DD.

According to matrix analysis, a Hermitian matrix is unitarily diagonalizable and the diagonal entries are necessarily the eigenvalues of it. The diagonalization can be achieved by a unitary transformation with the eigenvector matrix. Fortunately, the noise CSM $\boldsymbol{R}_n$ is Hermitian, so it can be diagonalized. The noise CSM $\boldsymbol{R}_n$ is normalized by $\sigma_n^2$, the mean power of its diagonal, and denoted by $\boldsymbol{\rho}_n$. Additionally, the eigendecomposition of $\boldsymbol{\rho}_n$ is expressed as:

$$\boldsymbol{\rho}_n = \boldsymbol{U} \boldsymbol{\Lambda} \boldsymbol{U}^{\mathrm{H}}, \tag{11}$$

where $\boldsymbol{U}$ is the eigenvector matrix and $\boldsymbol{\Lambda}$ is a diagonal matrix composed of eigenvalues. The eigenvector matrix is unitary, i.e., $\boldsymbol{U} \boldsymbol{U}^{\mathrm{H}} = \boldsymbol{I}$. The ambient noise is transformed by the unitary matrix $\boldsymbol{U}^{\mathrm{H}}$, $\boldsymbol{N}_d = \boldsymbol{U}^{\mathrm{H}} \boldsymbol{N}$, and the diagonalized CSM of noise is

$$\boldsymbol{R}_{dn} = \mathbb{E}(\boldsymbol{N}_d \boldsymbol{N}_d^{\mathrm{H}}) = \boldsymbol{U}^{\mathrm{H}} \boldsymbol{R}_n \boldsymbol{U} = \sigma_n^2 \boldsymbol{U}^{\mathrm{H}} \boldsymbol{\rho}_n \boldsymbol{U} = \sigma_n^2 \boldsymbol{\Lambda}. \tag{12}$$

Assuming that ambient noise is statistically stationary and $\boldsymbol{\rho}_n$ is measured before the source operates. In post-processing of the received data, the ambient noise CSM is used to implement diagonalization. The received data is transformed by the unitary matrix $\boldsymbol{U}^{\mathrm{H}}$, and the corresponding CSM of the transformed data is expressed as:

$$\boldsymbol{R}_d = \boldsymbol{U}^{\mathrm{H}} \boldsymbol{R} \boldsymbol{U} = \boldsymbol{U}^{\mathrm{H}} \boldsymbol{R}_s \boldsymbol{U} + \sigma_n^2 \boldsymbol{\Lambda}. \tag{13}$$

It is apparent that $\boldsymbol{R}_n$ is transformed into a diagonal matrix. In addition, it has been proven that a unitary transformation does not change the eigenvalues of $\boldsymbol{R}_s$ and also maintains the matrix to be positive semidefinite [24], so the DD algorithm can still be used to reduce the noise components in $\boldsymbol{R}_d$.

The denoised CSM $\boldsymbol{R}_{dDD}$ and the denoising matrix $\boldsymbol{D}_d$ for $\boldsymbol{R}_d$ are computed by Equation (9). To apply beamforming for the denoised CSM in the original signal space, the denoised CSM and the denoising matrix should be inversely transformed to the physical domain by

$$\boldsymbol{R}_{DD} = \boldsymbol{U}\boldsymbol{R}_{dDD}\boldsymbol{U}^{\mathrm{H}},$$
$$\boldsymbol{D} = \boldsymbol{U}\boldsymbol{D}_d\boldsymbol{U}^{\mathrm{H}}.$$

(14)

Then substituting the matrix $\boldsymbol{R}_{DD}$ into Equation (10), the PSD of the radiated noise is estimated. Thus, the influence of spatially correlated noise on radiated noise measurement is reduced.

For the proposed denoising method, temporally stationary ambient noise is crucial. If the ambient noise changes too much with time, the difference in the ambient noise CSM becomes larger before and after the measurement, which leads to a worse diagonalization and ultimately affects the denoising effect. Therefore, the premise of the proposed method is that the ambient noise should be statistically stationary. Under the ideal condition, the ambient noise CSM is the same as the received noise CSM, so the diagonalization for the received noise can be performed perfectly and the noise can be reduced well by DD. However, in practice, the number of snapshots is limited, so there is an inevitable mismatch between the ambient noise CSM and the received noise CSM because of the randomness of noise, which will result in imperfect diagonalization and thus decrease the denoising effect. In the next section, we will illustrate the denoising effect of the proposed DD-DD method for spatially correlated noise with limited snapshots via numerical simulations.

## 4. Simulations and Analysis

In this section, we consider a shallow water waveguide, and the environmental parameters and the arrangement of the source and the array are illustrated in Figure 1. A 15-element VLA with half-wavelength inter-element spacing is used to receive the radiated noise, and the center of the VLA is 50 m deep. The depth of the source is 10 m and the horizontal distance from the array is 100 m. The source signal is sampled from a normal distribution with $\sigma_s^2$ is 110 dB. The ambient noise is generated according to the correlation of the noise field, and the mean power of ambient noise $\sigma_n^2$ is 80 dB. The frequency of the signal is 750 Hz. In the following simulations, the effects of denoising for spatially correlated noise under the ideal condition and the realistic condition are verified firstly. Then, the influence of denoising on radiated noise measurement is analyzed with the SNR, the number of snapshots, and the frequency of the received data, respectively. Besides DD-DD, DD and WD-DD are included in the following simulations for comparison.

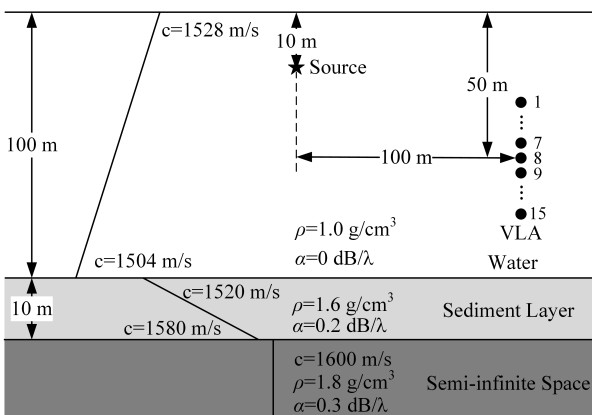

**Figure 1.** The sketch of the shallow water environment and the arrangement of the source and the VLA.

Considering that different hydrophones receive different signal powers and noise powers, the SNR of the received data is defined by the average powers of the received signal and noise

$$SNR = 10\lg \frac{\sigma_s^2 \frac{\boldsymbol{G}^{\mathrm{H}}\boldsymbol{G}}{M}}{\frac{1}{M}\sum_{m=1}^{M}\sigma_{n,m}^2} = 10\lg \frac{\sigma_s^2 \boldsymbol{G}^{\mathrm{H}}\boldsymbol{G}}{M\sigma_n^2},$$

(15)

where $\sigma_{n,m}^2$ denotes the noise power received by the $m$th hydrophone. In the above scenario, the average transmission loss, $-10\lg((\boldsymbol{G}^\mathrm{H}\boldsymbol{G})/M)$, is about 40 dB and the SNR of the received data is about $-10$ dB.

For the spatially correlated noise field, we consider the wind-generated noise and use the K/I model to calculate the vertical directionality and the correlation of it [25,26]. The vertical directionality and the normalized CSM $\boldsymbol{\rho}_n$ of the wind-generated noise at 750 Hz in shallow water are shown in Figure 2.

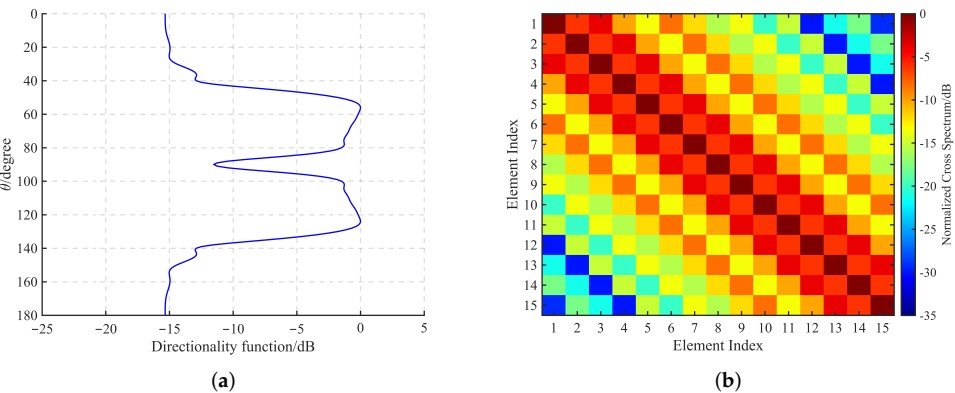

(a)　　　　　　　　(b)

**Figure 2.** (**a**) The vertical directionality and (**b**) the normalized CSM of the wind-generated noise at 750 Hz in shallow water.

To generate noise with the specific correlation in simulations, the following approach is applied. First, the normalized noise CSM is decomposed by the Cholesky decomposition as $\boldsymbol{\rho}_n = \boldsymbol{B}\boldsymbol{B}^\mathrm{H}$. Then, the correlated noise is constructed by left multiplying the white Gaussian noise $\boldsymbol{N}$ by the matrix $\boldsymbol{B}$, and CSM of the constructed noise is expressed as:

$$\boldsymbol{R}_n = \mathbb{E}((\boldsymbol{B}\boldsymbol{N})(\boldsymbol{B}\boldsymbol{N})^\mathrm{H}) = \boldsymbol{B}\mathbb{E}(\boldsymbol{N}\boldsymbol{N}^\mathrm{H})\boldsymbol{B}^\mathrm{H} = \boldsymbol{B}\sigma_n^2\boldsymbol{I}\boldsymbol{B}^\mathrm{H} = \sigma_n^2\boldsymbol{\rho}_n, \tag{16}$$

where $\boldsymbol{I}$ denotes the identity matrix. If the number of snapshots is limited, the CSM of white Gaussian noise is not an identity matrix, resulting in the difference between the constructed noise CSM and the ideal wind-generated noise CSM.

*4.1. Noise Reduction Analysis for Spatially Correlated Noise*

First, we consider the ideal condition that the number of snapshots is infinite. The received noise CSM constructed by Equation (16) is the same as the ambient noise CSM. The received noise CSM, and its whitening matrix and diagonalizing matrix are shown in Figure 3. It is clear that the whitening and diagonalization are perfect, and the received noise CSM is transformed into an identity matrix multiplied by $\sigma_n^2$ and a diagonal matrix, respectively.

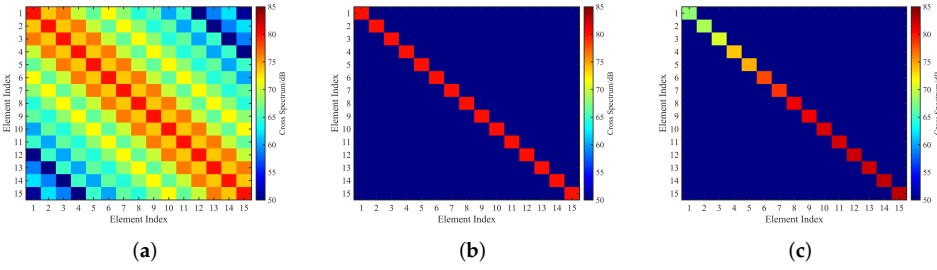

(a)　　　　　　　(b)　　　　　　　(c)

**Figure 3.** (**a**) The received noise CSM, its (**b**) whitening matrix, and (**c**) diagonalizing matrix under the ideal condition that assumes that the ambient noise CSM is the same as the received noise CSM.

The denoising factors calculated by DD for the received noise CSM, the whitened CSM, and diagonalized CSM are shown in Figure 4. It can be seen from Figure 4b,c that the denoising factors are equal to the optimal values, which means the denoising effects of DD-DD and WD-DD are equal and better than that of DD shown in Figure 4a under the ideal condition.

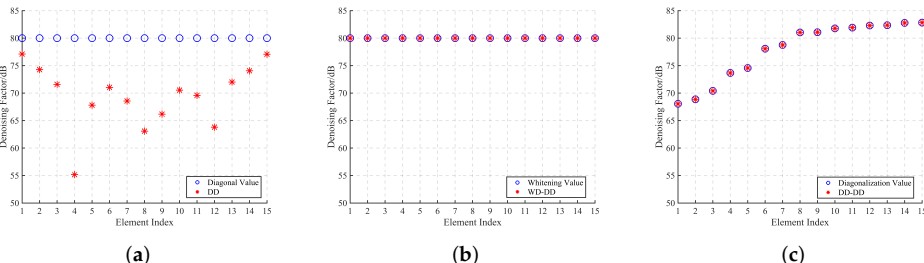

(a)                      (b)                      (c)

**Figure 4.** The diagonal elements and the calculated denoising factors for (**a**) the received noise CSM, (**b**) the whitened CSM, and (**c**) the diagonalized CSM under the ideal condition.

For the realistic condition, assuming the number of snapshots is 50, the array received noise is constructed by the normalized CSM of the wind-generated noise. The CSM of the received noise $\boldsymbol{R}_n$ is shown in Figure 5a. Comparing Figure 5a with Figure 3a, there is an evident difference because of the randomness of noise when the number of snapshots is limited. Hence, the whitening and diagonalization of the received noise CSM by $\boldsymbol{\rho}_n$ are imperfect, resulting in two approximately diagonal matrices shown in Figure 5b,c.

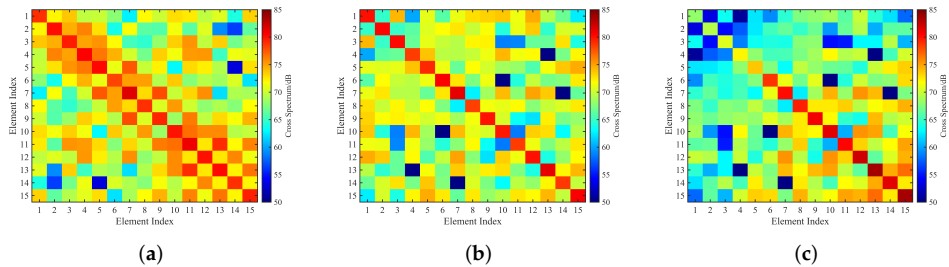

(a)                      (b)                      (c)

**Figure 5.** (**a**) The CSM of the received noise with 50 snapshots and its (**b**) whitening matrix, and (**c**) diagonalizing matrix.

The denoising factors for Figure 5a–c is displayed in Figure 6a–c, respectively. The denoising matrices calculated by DD, WD-DD, and DD-DD in the physical domain are illustrated in Figure 6d–f, respectively. It can be seen from the results that all three denoising methods can reduce the noise to different degrees. However, DD only reduces the diagonal elements, and the denoising factors are much smaller than the diagonal elements of the received noise CSM as illustrated in Figure 6a, resulting in the poor denoising effect for spatially correlated noise. According to Figure 6e,f, WD-DD and DD-DD reduce not only the diagonal elements but also the off-diagonal elements. In addition, more denoising factors of DD-DD are close to the diagonal elements of the diagonalized noise CSM and thus the denoising matrix of DD-DD is closer to the received noise CSM, seen in Figures 5a and 6f. All results illustrate that the performance of DD-DD is better than the other two denoising methods for spatially correlated noise.

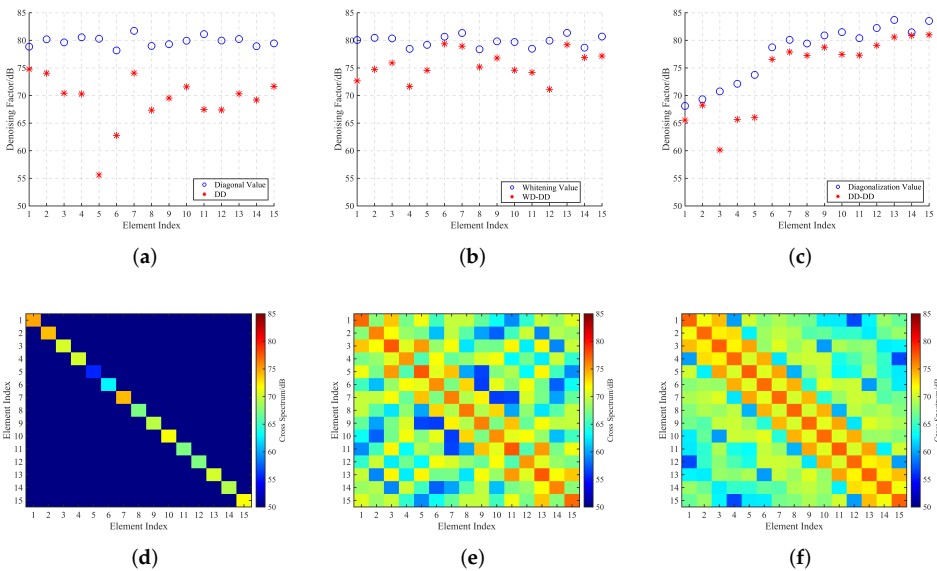

**Figure 6.** The denoising results by DD, WD-DD, and DD-DD. (**a**–**c**) show the denoising factors for the received noise CSM and its whitening and diagonalization, (**d**–**f**) show the denoising matrices in the physical domain calculated by DD, WD-DD, and DD-DD, respectively.

*4.2. The Influence of Denoising on Underwater Radiated Noise Measurement*

For underwater radiated noise measurement, the data is sampled in the time domain with a 10 kHz sampling frequency. The time-domain signal segment is 5 s for a single processing operation, and the number of snapshots in the frequency-domain is 49 by STFT using a 0.2 s rectangular window with 50% overlap. In the following simulations, the SNR, the number of snapshots, and the frequency of the received data are changed separately to illustrate its influence on underwater radiated noise measurement.

To quantify the error of the estimated PSD and the real value, the Root-Mean-Square Error (RMSE) of the estimated PSD is defined as

$$\text{RMSE} = \sqrt{\mathbb{E}(\hat{P}_S - P_S)^2}. \tag{17}$$

The expectation in the formula can be approximated by averaging multiple independent results. The RMSEs in the following simulations are calculated by 500 independent trials.

The variation of RMSE with the SNR of the received data is shown in Figure 7. The power of the radiated noise increases from 100 dB to 140 dB, and the corresponding SNR range of the received data is about $[-20, 20]$ dB because of the 40 dB transmission loss. It can be seen that all RMSEs decline with the increasing SNR. The denoising effects on radiated noise measurement with WD-DD and DD-DD are apparent when the SNR of the received signal is under 5 dB, while the denoising effect of DD for spatially correlated noise is not evident. The RMSE of the radiated noise measurement with DD-DD is about 1.6 dB smaller than the RMSE without denoising when the SNR of the received data is under $-10$ dB. When the SNR is over 10 dB, the RMSE of noise measurement is tiny and the performance will hardly be improved by the denoising technique.

Figure 8 shows the variation of the RMSE of radiated noise measurement with the number of snapshots. The RMSEs of DD and no-denoising decrease with the increasing snapshots, but the downtrend becomes tiny when the number of snapshots exceeds 30, which is 2 times the number of hydrophones. Snapshot number is not the critical factor for DD to reduce spatially correlated noise, but the correlation of noise is. The improvements of WD-DD and DD-DD on radiated noise measurement arise when the snapshot number is greater than the number of hydrophones. The RMSEs of WD-DD and DD-DD decrease rapidly with the increasing number of snapshots. Moreover, the RMSE with DD-DD is smaller than the other two denoising methods.

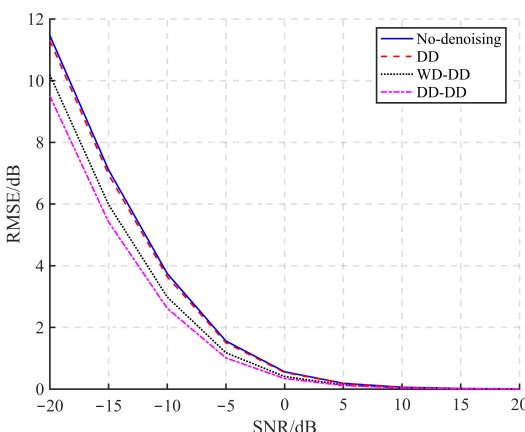

**Figure 7.** The variation of RMSE with the SNR of the received data.

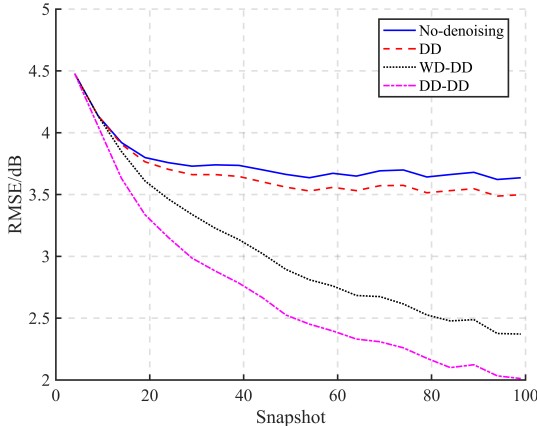

**Figure 8.** The variation of RMSE with the number of snapshots.

In practice, an array with fixed sensor spacing is usually used to measure the broadband radiated noise, so the sensor spacing does not satisfy the requirement of half-wavelength for most frequencies. Assuming that the sensor spacing is 1 m, and the RMSE varying within the frequency band 500–1000 Hz is shown in Figure 9.

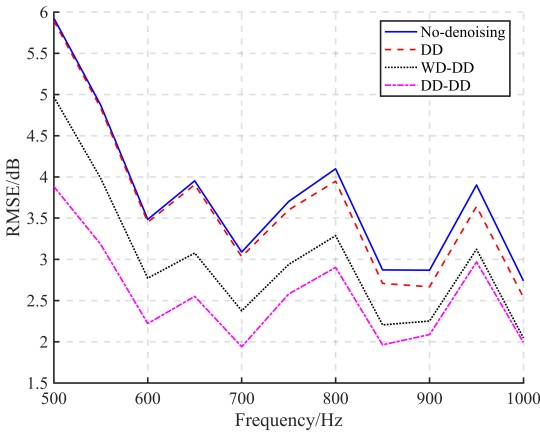

**Figure 9.** The variation of RMSE with the frequency.

The spatial correlation distance of the ambient noise decreases with the increasing frequency, so the correlation of the noise received by an array with the fixed sensor spacing decreases, which results in a more diagonal noise CSM. That is the reason why the im-

provement of DD on radiated noise measurement becomes significant with the increasing frequency, as illustrated in Figure 9. In addition, the denoising effects on noise measurement with WD-DD and DD-DD are evident, while the RMSE of DD-DD is minimal in the simulation frequency band.

## 5. Experiment Validation

In this section, the performance of DD-DD on underwater radiated noise measurement is validated with experimental data. The experiment was implemented in a reservoir in December 2019. The environment of the reservoir and the arrangement of the source and the array are shown in Figure 10. The sound speed profile in the water is measured by a CTD (standing for "Conductivity, Temperature, and Depth"). The speeds, densities and attenuations of the sound waves in the sediment layer and semi-infinite space are obtained according to the bottom material. A transducer (UW-350) located at the depth of 50 m was used as a source to transmit two types of signal, a Continuous Wave (CW) signal with the frequency of 750 Hz and a Linear Frequency Modulation (LFM) signal with the frequency band of 100–900 Hz and the duration 10 s. A nested array (consisting of a 24-element uniform linear array with 1 m inter-element spacing and a 17-element uniform linear array with 0.5 m inter-element spacing) was vertically deployed at 73–97 m deep, and 34.5 m away from the source. In addition, a single hydrophone located at the same depth as the source and a horizontal distance of 5.2 m away from it was used to monitor the radiated signals. The SNR of the hydrophone received data is high because of the small transmission distance, thus the influence of noise can be neglected and the PSD estimated by the hydrophone is regarded as the true value of the radiated signal. All data was recorded at a 48 kHz sampling frequency.

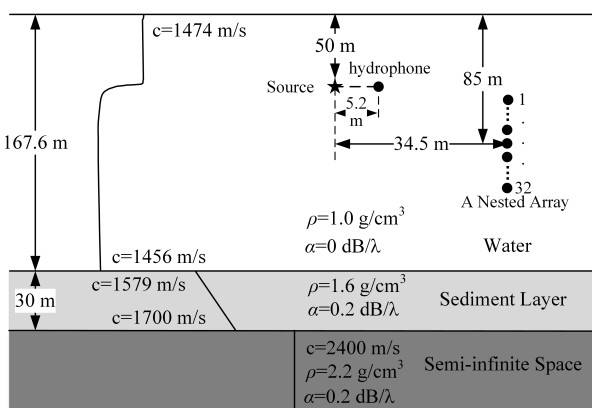

**Figure 10.** The sketch of the experimental reservoir environment and the arrangement of the source and the array.

In post-processing, we select the data of the central 15 hydrophones of the nested array with 1 m inter-element spacing. To illustrate the influence of snapshot number on the measurement results, the length of the CW signal is changed from 2 s to 10 s, and the data is transformed into the frequency-domain by STFT using a 0.2 s rectangular window with 50% overlap. The PSD of the CW signal is estimated with different numbers of snapshots, shown in Figure 11a, and the corresponding estimation errors are shown in Figure 11b, which are basically consistent with the simulation results shown in Figure 8. It can be seen that the estimation error shows a decreasing trend with the increase of the snapshot number, and the downtrend gradually becomes undulating when the snapshot number exceeds 50. The estimated PSD with DD is almost the same as that without denoising, which indicates that the denoising effect of DD is negligible. The denoising effect of WD-DD is little or even worse for small snapshot numbers, but it becomes apparent when the number of snapshots is more than 50. The reason for this result is that the mismatch between the ambient noise CSM and the received noise CSM is severe when the snapshot number is small, while whitening is sensitive to the mismatch. In addition, the DD-DD has good

results at small snapshot numbers and the improvement of DD-DD on underwater radiated noise measurement is best compared with DD and WD-DD.

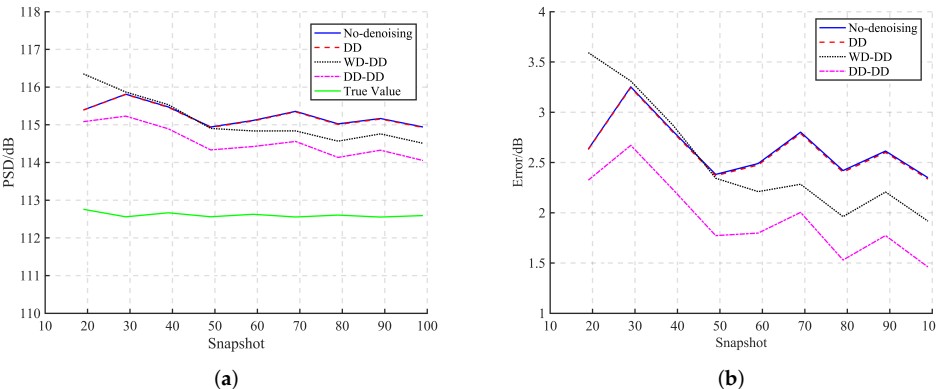

**Figure 11.** (**a**) The PSDs and (**b**) estimation errors of the CW signal with different numbers of snapshots.

To compare the measurement results at different frequencies, the PSDs and estimation errors for a 10 s LFM signal within 500–900 Hz are estimated with no-denoising, DD, WD-DD, and DD-DD, respectively, and the results are shown in Figure 12. As shown in the figures, all denoising methods reduce the estimation error at different frequencies except DD. The PSD estimated by DD is almost identical to the result without denoising, which has little effect on the improvement of the accuracy of radiated noise measurement. Moreover, both WD-DD and DD-DD can reduce the influence of ambient noise and improve the accuracy of PSD estimation, but DD-DD has the best performance on underwater radiated noise measurement.

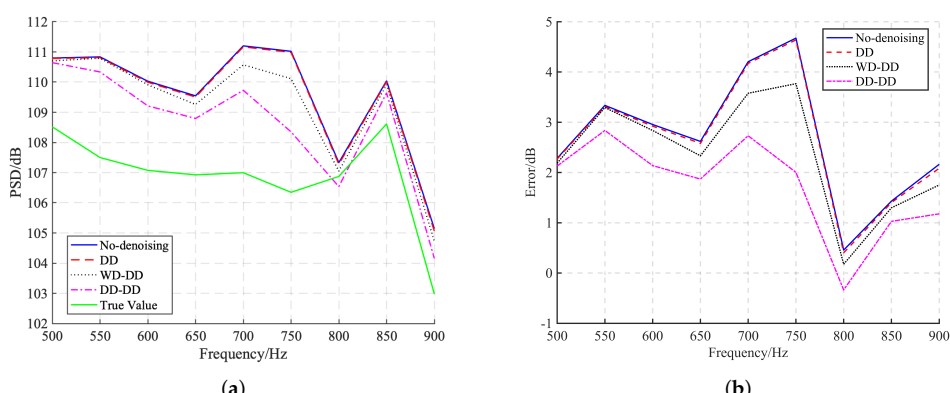

**Figure 12.** (**a**) The PSDs and (**b**) estimation errors of the LFM signal with different frequencies.

## 6. Conclusions

The diagonalization-decorrelation-based diagonal denoising method is proposed to improve the denoising effect for spatially correlated noise in underwater radiated noise measurement. In this method, the eigenvector matrix of the normalized ambient noise CSM is utilized to diagonalize the received data CSM by unitary transformation, which transforms the received noise CSM into a diagonal matrix consisting of its eigenvalues. The DD technique is applied subsequently to reduce the noise components. The denoised CSM instead of the received data CSM is used to estimate the PSD of the radiated noise by beamforming, which decreases the influence of ambient noise and improves the accuracy of radiated noise measurement. The diagonalized noise CSM is an ideal diagonal matrix under the ideal condition that the ambient noise CSM is identical to the received noise

CSM, and the noise components can be removed entirely by DD. In a realistic case, the number of snapshots is limited, so there is an inevitable mismatch between the ambient noise CSM and the received noise CSM due to the randomness of noise. Therefore, the diagonalized noise CSM is not an ideal diagonal matrix, resulting in the noise reduction effect decreasing. The denoising effect is verified via numerical simulations under ideal and realistic conditions. The simulation results show that the proposed DD-DD method still has a good noise reduction effect for spatially correlated noise compared with DD and WD-DD even if the number of snapshots is limited. Meanwhile, the radiated noise measurement accuracy is improved evidently by DD-DD under different conditions with the SNR of received data, the number of snapshots, and the frequency changing separately. Finally, the validity of DD-DD on radiated noise measurement in practice is further verified by experimental data.

**Author Contributions:** Conceptualization, G.J.; methodology, G.J., C.S. and L.X.; validation, G.J.; formal analysis, G.J.; investigation, G.J. and L.X.; resources, C.S.; data curation, G.J.; writing—original draft preparation, G.J.; writing—review and editing, C.S. and L.X.; supervision, C.S. and L.X.; project administration, C.S.; funding acquisition, C.S. All authors have read and agreed to the published version of the manuscript.

**Funding:** This research was supported by the National Natural Science Foundation of China under Grant 11974285.

**Institutional Review Board Statement:** Not applicable.

**Informed Consent Statement:** Not applicable.

**Data Availability Statement:** The data presented in this study are available on request from the corresponding author.

**Conflicts of Interest:** The authors declare no conflict of interest.

## Appendix A

Considering the ideal CSM of the received data expressed as Equation (8) with spatially uncorrelated noise, the signal CSM is positive semidefinite, and there must be a vector $\exists v \in \ker(\boldsymbol{R}_s)$ such that $\boldsymbol{v}^H \boldsymbol{R}_s \boldsymbol{v} = 0$. So

$$\begin{aligned} \boldsymbol{v}^H \boldsymbol{R} \boldsymbol{v} &= \boldsymbol{v}^H (\boldsymbol{R}_s + \boldsymbol{R}_n) \boldsymbol{v} \\ &= \boldsymbol{v}^H (\boldsymbol{R}_{DD} + \boldsymbol{D}) \boldsymbol{v}, \end{aligned} \tag{A1}$$

$$\begin{aligned} \boldsymbol{v}^H \boldsymbol{R}_{DD} \boldsymbol{v} &= \boldsymbol{v}^H (\boldsymbol{R}_n - \boldsymbol{D}) \boldsymbol{v} \\ &= \sum_{m=1}^{M} (\sigma_{n,m}^2 - d_m) |v_m|^2 \geq 0, \end{aligned} \tag{A2}$$

where $\sigma_{n,m}^2$ and $d_m$ is the $m$th diagonal element of the received noise CSM $\boldsymbol{R}_n$ and the denoising matrix $\boldsymbol{D}$, respectively. The formula $\boldsymbol{v}^H \boldsymbol{R}_{DD} \boldsymbol{v} \geq 0$ holds because the matrix $\boldsymbol{R}_{DD}$ is constrained to be positive semidefinite in the DD algorithm. In the maximum of $\boldsymbol{d}$, the result does not exceed the diagonal elements of $\boldsymbol{R}_n$ in the direction $(|v_1|^2, |v_2|^2, \ldots, |v_M|^2)^T$. Hence, DD does not reduce the power of the received signal.

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
