# Peer review of "Diagonal Denoising for Spatially Correlated Noise Based on Diagonalization Decorrelation in Underwater Radiated Noise Measurement"

_jmse, doi:10.3390/jmse10040502_

Round 1

Reviewer 1 Report

Please see the review report attached. 

Reviewer 2 Report

ABOUT ACRONYMS
· When you include one acronym, write the first letter of the definition in capital letters, then you can use only the acronym. Examples: "Sound Pressure Level (SPL)" [line 24]. Please, review the article to correct this mistake
· Try not to include acronyms in the abstract
· It uses the acronym TL commonly understood as Transmission Loss but does not describe it anywhere and uses it in some equations as a parameter.
· Check the acronyms used, I think you have abused them. For example in the case of CBF [line 55], it is Conventional Beamforming (CB) or Conventional BeanForming (CBF). Acronyms should help the reader to read quickly, if you abuse them, even adding acronyms that are only used once (e.g. AT in line 137), the reader gets lost very easily.

ABOUT THE BIBLIOGRAPHY
· In support of the claim "Underwater radiated noise measurement is essential to monitor the anthropogenic noise in the ocean" [lines 24-25], believes it necessary to reference all its citations 2-7?
· In support of the claim "The common radiated noise measurement approach is to use a single hydrophone" [lines 27-28], believes it necessary to reference all its citations 8-12?
· Much of the bibliography used seems forced to appear, you should really check and include only those citations that support your text and your work, more is not better.
· Do you really think it is even necessary to cite matrix analysis? [line 183]
· In my opinion, it is wrong to cite in this way: "In [x]..." (e.g. in line 38 and 39), or "The author X [x]" (e.g. in lines 53-54, 59), or "The ... method [20,21] can..." (line 49). Please, read more quality papers and learn that cites are at the end of the sentence that refers to the cite (e.g. in line 24). The objective of a paper is to explain (and include) the idea of the work in the text, do not force the reader to read all the references you have included. The citations exist to demonstrate that you are not inventing anything, and you are getting ideas from other sources that you assumed were true.

ABOUT THE TEXT
· Use capital letters on the Keyboards
· Do you really consider it necessary to repeat equation 9 as you have done with equation 14?
· Regarding Figure 1: indicate which element of the array would be element 1 and which element would be element n in order to know the correct enumeration, and indicate whether this is relevant for your model.
· It would be good if you could repeat Figure 1 but conditioned to your particular experiment, indicating all kinds of values used. 

ABOUT THE WORK:
· Can the space between hydrophones in the array affecting to the model? What would happen if they were not equidistant from each other? Comment on the case in the text to clarify it and show that you have everything under control.
